# Vocal Sandbox: Continual Learning and Adaptation for Situated Human-Robot Collaboration

**Jennifer Grannen*, Siddharth Karamcheti*, Suvir Mirchandani, Percy Liang, Dorsa Sadigh**
Stanford University, Stanford, CA
https://vocal-sandbox.github.io

**Abstract:** We introduce Vocal Sandbox, a framework for enabling seamless human-robot collaboration in situated environments. Systems in our framework are characterized by their ability to *adapt and continually learn* at multiple levels of abstraction from diverse teaching modalities such as spoken dialogue, object keypoints, and kinesthetic demonstrations. To enable such adaptation, we design lightweight and interpretable learning algorithms that allow users to build an understanding and co-adapt to a robot's capabilities in real-time, as they teach new behaviors. For example, after demonstrating a new low-level skill for "tracking around" an object, users are provided with trajectory visualizations of the robot's intended motion when asked to track a new object. Similarly, users teach high-level planning behaviors through spoken dialogue, using pretrained language models to synthesize behaviors such as "packing an object away" as compositions of low-level skills – concepts that can be reused and built upon. We evaluate Vocal Sandbox in two settings: collaborative gift bag assembly and LEGO stop-motion animation. In the first setting, we run systematic ablations and user studies with 8 non-expert participants, highlighting the impact of multi-level teaching. Across 23 hours of total robot interaction time, users teach 17 new high-level behaviors with an average of 16 novel low-level skills, requiring 22.1% less active supervision compared to baselines. Qualitatively, users strongly prefer Vocal Sandbox systems due to their ease of use (+31.2%), helpfulness (+13.0%), and overall performance (+18.2%). Finally, we pair an experienced system-user with a robot to film a stop-motion animation; over two hours of continuous collaboration, the user teaches progressively more complex motion skills to produce a 52 second (232 frame) movie.

**Keywords:** Continual Learning, Multimodal Teaching, Human-Robot Interaction

## 1  Introduction

Effective human-robot collaboration requires systems that can seamlessly *work with* and *learn from* people [1–3]. This is especially important for situated interactions [4–7] where robots and people share the same space, working together to achieve complex goals. Such settings require robots that continually adapt from diverse feedback, learning and grounding new concepts online. Yet, recent work [8–15] remain limited in the types of teaching and generalization they permit. For example, many systems use language models to map user utterances to sequences of skills from a static, predefined library [16–18]. While these systems may generalize at the plan level, they trivially fail when asked to execute new low-level skills – regardless of how complex that skill might be (e.g., "hold this still"). Instead, we argue that situated human-robot collaboration requires learning and adaptation at *multiple levels of abstraction*, empowering collaborators to continuously teach new high-level planning behaviors *and* low-level skills over the course of an interaction.

Building on this insight, we present **Vocal Sandbox**, a framework for situated human-robot collaboration that enables users to teach via diverse modalities such as spoken dialogue, object keypoints, and kinesthetic demonstrations. Systems in our framework consist of a language model (LM) planner [19] that maps user utterances to sequences of high-level behaviors, and a family of low-level skills that

---

*Equal Contribution. Correspondence to {jgrannen,skaramcheti}@stanford.edu.

8th Conference on Robot Learning (CoRL 2024), Munich, Germany.

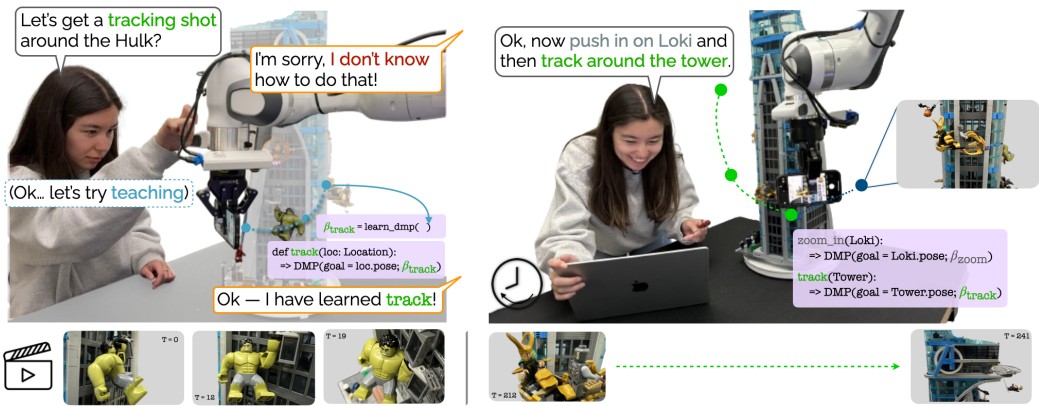

Figure 1: **Motivating Example**. We present Vocal Sandbox, a framework for human-robot collaboration that enables robots to *adapt and continually learn* from situated interactions. In this example, a user arranges individual LEGO structures for each frame of a stop-motion film **[Bottom]**, while a robot arm controls the camera. The user teaches the robot new behaviors through feedback modalities such as language and demonstrations **[Left]**. The robot learns *online*, scaling to more complex tasks as the collaboration continues **[Right]**.

ground each behavior to robot actions. Crucially, we design lightweight and interpretable learning algorithms to incorporate a user's teaching feedback, *dynamically growing the capabilities of the system in real-time*. Consider a user and robot collaborating to film a LEGO stop-motion animation (Fig. 1).[1] The user is the animator, articulating LEGO figures and structures to rig each keyframe (Fig. 1; Bottom), while the robot carries out precise camera motions (e.g., tracking smoothly to zoom into a character). Early in the interaction, the user asks: "Let's get a tracking shot around the Hulk?" In response, the LM attempts to generate a plan, but *fails* – "tracking" is not a known concept. The robot vocalizes its failure, deferring to the user for next steps. Empowered by our framework, the user chooses to *explicitly teach* the robot how to "track" by providing a kinesthetic demonstration. Using this single example as supervision (Fig. 1; Left), we immediately parameterize a new skill (i.e., $\beta_{\text{track}}$), and synthesize the corresponding behavior (i.e., track(loc: Location)). Later (Fig. 1; Right), when the user directs to robot to "push in on Loki and then track around the tower," the robot can immediately use what it has learned to generate both the full task plan (zoom_in(Loki); track(Tower)), as well as a visualization of the motion trajectory (shown via a custom interface; §3.3). On user confirmation, the robot executes in the real-world, capturing the desired shot.

We evaluate Vocal Sandbox through systematic ablations and long-horizon user studies. In our first setting, non-expert participants work with a Franka Emika fixed-arm manipulator for the task of collaborative gift bag assembly; we run a within-subjects user study with $N = 8$ users spanning a total of 23 hours of robot interaction time, comparing a Vocal Sandbox system with two baselines: a static variant of our system without learning at either the high- and low-level [8], as well as a variant of our system with only high-level teaching. We show that across all three methods, our contributions enable users to teach 17 new high-level behaviors and an average of 16 new low-level skills, resulting in 22.1% faster collaborative task performance compared to baselines; qualitatively, participants strongly prefer our system over all baselines due to its ease of use (+29.4%), helpfulness (+10.8%) and overall performance (+13.9%). We then scale to a more advanced setting where an experienced system-user (an author of this work) works with a robot to shoot a stop-motion animation (Fig. 1); over *two hours of continuous collaboration* the user teaches progressively more complex concepts, building a rich library of skills and behaviors to produce a 52 second (232 frame) movie.

## 2 Vocal Sandbox: A Framework for Continual Learning and Adaptation

Vocal Sandbox is characterized by a *language model planner* that maps user utterances to sequences of high-level behaviors and a *family of skill policies* that ground behaviors to robot actions (Fig. 2). In this section, we first formalize planning (§2.1) and skill execution (§2.2), then introduce our contributions for teaching new behaviors from diverse modes of feedback (§2.3).

---

[1]Wikipedia provides a thorough overview of the rich history and different styles of stop-motion animation, with a dedicated page on "Brickfilm" (e.g., using LEGOs) – the sub-genre we focus on in this work.

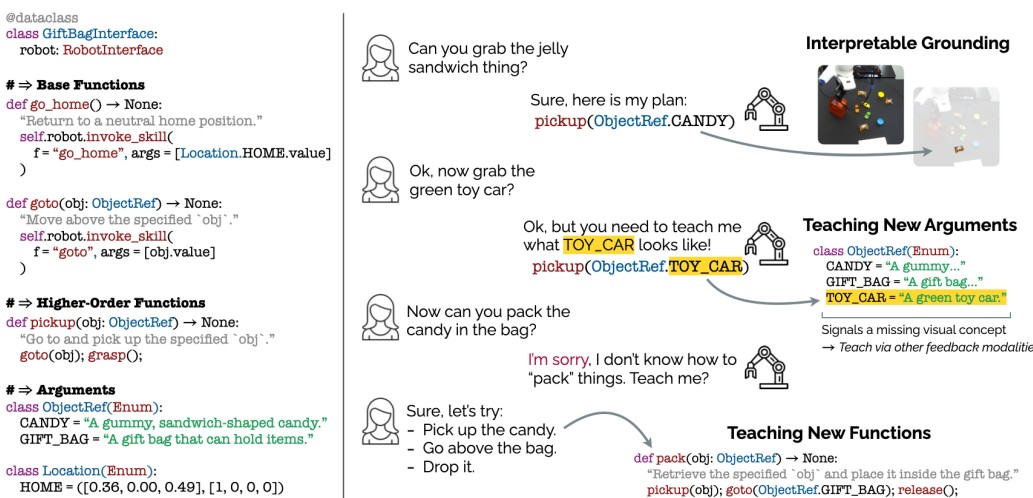

Figure 2: **Planning & Teaching with Language Models**. We use language models to parse user utterances to plans – executable programs with functions and arguments defined by an API **[Left]**. Given a successful parse, we visualize an *interpretable trace* of both the plan and robot's intended behavior on a custom GUI (§3.3). In the case of failure, we elicit teaching feedback from users to *synthesize* new functions and arguments (§2.3).

## 2.1 High-Level Planning with Language Models

We implement high-level planning using a language model (LM) prompted with an API specification $\Lambda_t = (\mathcal{F}, \mathcal{C})$ (Fig. 2; Left) that consists of a set of functions $\mathcal{F}$ (synonymous with "behaviors") and argument literals $\mathcal{C}$, indexed by interaction timestep $t$ – this indexing is important as users can teach new behaviors (add to the API) over the course of a collaboration. Each argument $c \in \mathcal{C}$ is a typed literal – for example, we define Enums such as Location that associate a canonical name (e.g., Location.HOME in Fig. 2) to values that are used by the robot when executing a skill. We define each function $f \in \mathcal{F}$ as a tuple $(n, \sigma, d, b)$ consisting of a semantically meaningful name $n$ (e.g., goto), a type signature $\sigma$ (e.g., [obj: ObjectRef] → None), human-readable descriptive docstring $d$ (e.g., "Move above the specified obj"), and function body $b$. Crucially, we assume we can define *higher-order* functions – for example pickup(obj: ObjectRef) as goto(obj); grasp().

Given a user utterance at time $t$, the language model generates a plan $p_t$ that is comprised of a program, or sequence of function invocations. For example, a valid plan for an instruction such as "place the candy in the gift bag" subject to the API in Fig. 2 would be $p_t$ = pickup(ObjectRef.CANDY); goto(ObjectRef.GIFT_BAG); release(). We formalize planning as generating $p_t = \text{LM}(\cdot \mid u_t, \Lambda_t, h_t)$, where $u_t$ is the user's utterance, $\Lambda_t$ is the current API, and $h_t$ corresponds to the full interaction history through $t$. Concretely, $h_t = (u_{\text{prompt}}, [(u_1, p_1), (u_2, p_2), \ldots (u_{t-1}, p_{t-1})])$; $u_{\text{prompt}}$ is the *system prompt* that describes the setting and constraints (e.g., "return plans as well-formed Python"; see our project page for full prompts).

Note that there are cases where the language model *may not be able to generate a valid plan* – for example, given an utterance describing a behavior that does not exist in the API (e.g., "can you pack a gift bag with three candies"). In such a situation, we rely on the commonsense understanding of the LM to generate a helpful error message that indicates the failure mode (e.g., "I am not sure how to *pack*; could you teach me?"). These error messages (as well as the successfully generated plans) are shown to users via a custom graphical interface (GUI; §3.3) to inform their next action.

## 2.2 Low-Level Skills for Action Execution

To ground plans $p_t$ to robot actions, we assemble skill policies $\pi_f(a \mid o_t, [\![c_1, c_2, \ldots]\!])$ that define the execution of a function $f \in \mathcal{F}$ as a mapping from a state observation $o_t \in \mathbb{R}^n$ and individual "resolved" arguments $[\![c_1, c_2, \ldots]\!]$ to a sequence of robot actions $a \in \mathbb{R}^{T \times D}$ (e.g., end-effector poses). Of particular note is the *resolution operator* $[\![\cdot]\!]$ that maps an LM-generated plan $p_t$ to a sequence of skill policy invocations; to do this, we first iterate through each function invocation in $p_t$ and recursively expand higher-order functions as necessary. In our "place the candy in the gift bag"

example, this means expanding the initial `pickup(ObjectRef.Candy)` to its resolved function body $[\![b_{\text{pickup}}]\!]$ `goto(ObjectRef.Candy)` and `release()`. We then resolve each *argument*, substituting each literal (variable name) with its corresponding value – for example $[\![$`ObjectRef.Candy`$]\!] =$ "A gummy, sandwich-shaped candy" while $[\![$`Location.HOME`$]\!] = ([0.36, 0.00, 0.49]_{\text{pos}}, [1, 0, 0, 0]_{\text{rot}})$. The corresponding values are passed to the underlying policy implementation $\pi_f$.

In this work, we look at three different classes of skill policies: hand-coded primitives (e.g., `GO_HOME` or `GRASP`), visual keypoint-conditioned policies that identify end-effector poses given natural language object descriptions, and dynamic movement primitives (§3.2). Note that in general, our formulation permits arbitrary policy classes, including learned closed-loop control policies [20–22].

### 2.3   Teaching via Program Synthesis

A core component of Vocal Sandbox is the ability to *synthesize* new behaviors and skills from diverse modalities of teaching feedback such as spoken dialogue, object keypoints, and kinesthetic demonstrations. To do this, we leverage the commonsense priors and strong generalization ability of our underlying language model to synthesize new functions and arguments, updating the API $\Lambda_t$ in real-time. Given an failed plan, each "teaching" step first uses the language model to vocalize the "missing" concept(s) to be taught, followed by a interaction that prompts the user to provide targeted feedback. The language model then *synthesizes* new functions and arguments for the API. Fig. 2 (Right) shows the two types of teaching we develop: 1) *argument teaching* and 2) *function teaching*.

The goal of *argument teaching* is to teach and ground new literals $\hat{c} \in \mathcal{C}$ – for example, identifying "green toy car" as a new argument given the utterance "Grab the green toy car" in Fig. 2 (Right). To do this, the LM parses as much of the utterance as possible subject to the current API, mapping "grab" to `pickup`. Because it is able to identify the correct function (but not the arguments), the LM then uses the corresponding type signature to *infer* that "green toy car" should be of type `ObjectRef`; it then *automatically synthesizes* an API update, adding a new literal `ObjectRef.TOY_CAR`. This addition is then shown to the user (as part of the second stage of teaching), who then then provides the supervision needed to successfully ground the literal (i.e., providing the keypoint supervision to localize the object). On successful execution, the LM commits this change, yielding $\Lambda_{t+1}$.

The goal of *function teaching* is to teach new functions $\hat{f} \in \mathcal{F}$ – for example, defining `pack` from the utterance "now can you pack the candy in the bag" in Fig. 2 (Right). Here, the LM cannot partially parse the utterance – "pack" does not have an associated function, so there is no reliable way to infer a type signature. Instead, the LM highlights "pack" as a new behavior, and explicitly asks the user to teach its meaning through decomposition [23, 24], breaking "pack" down into a chain of existing skills. In this case, the user says: "Pick up the candy; go above the bag; drop it" with program `pickup(ObjectRef.CANDY); goto(GIFT_BAG); release()`. The LM then *explicitly synthesizes* the new function $\hat{f} = (\hat{n}, \hat{\sigma}, \hat{d})$, with name $\hat{n} =$ `pack`, signature $\hat{\sigma} =$ `obj: ObjectRef → None`, and docstring $\hat{d} =$ "Retrieve the object and place it in the gift bag." We also generate the "lifted" body `pickup(obj); goto(GIFT_BAG); release()` via first-order abstraction [24, 25].

This combination of argument and function teaching enables the expressivity and real-time adaptivity of our framework; in defining lightweight algorithms for learning and synthesizing new API specifications from interaction, we provide users with a reliable method of growing and reasoning over the robot's capabilities during the course of a collaboration.

## 3   Implementation & Design Decisions

While we introduce Vocal Sandbox as a general learning framework, this section provides implementation details specific to the experimental settings we explore in our experiments. Notably, our first experimental setting involves a collaborative gift-bag assembly task (§4.1) where a non-expert user and robot work together to assemble a gift bag with a set of known objects (visualized in Fig. 2). Our second setting (§4.2) pairs an experienced system user (an author of this work) with a robot for the task of LEGO stop-motion animation (visualized in Fig. 1). For all settings, we use a Franka Emika Panda arm equipped with a Robotiq 2F-85 gripper following DROID [26]; we also assume access to an overhead ZED 2 RGB-D camera with known intrinsics and extrinsics to obtain point clouds.

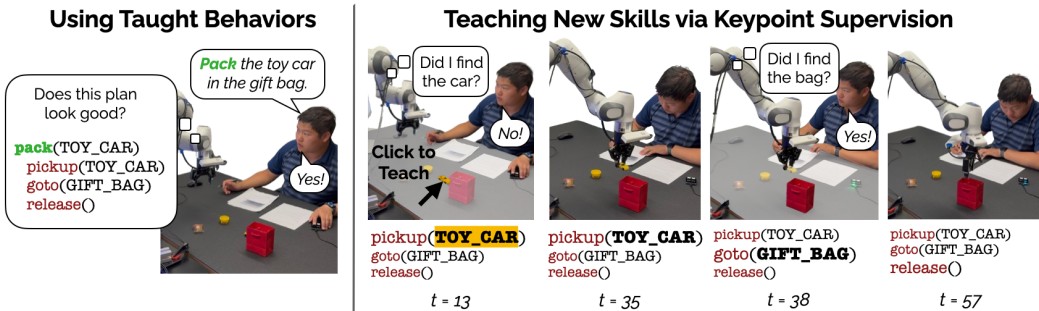

Figure 3: **Collaborative Gift-Bag Assembly**. In this example, a study participant (§4.1) verbally asks the robot to "pack the toy car in the gift bag," leveraging pack, a newly taught behavior, to minimize his time supervising **[Left]**. When the robot fails to localize the "car" in the image, the user corrects this by *clicking* on the interactive GUI, producing a keypoint label, teaching a new argument and grounding corresponding skill **[Right]**.

### 3.1 Language Models for High-Level Planning

We use GPT-3.5 Turbo with function calling [v11-06; 27, 28] due to its high latency and cost-effective pricing. We encode $\Lambda_t$ as a Python code block (formatted as Markdown) in the system prompt $u_{\text{prompt}}$. To constrain the LM outputs to well-formed programs, we use the function calling feature provided by the OpenAI chat completion endpoint [28], formatting each function in $\Lambda_t$ as a "tool" that the LM can invoke in response to a user endpoint. Full code and prompts are on our project page.

### 3.2 Skill Policies for Object Manipulation and Dynamic Motions

We implement different families of skill policies $\pi_f$ for each of our two experimental settings. In our first setting (§4.1) we implement skills using a visual keypoints model, while for our second setting (§4.2) we implement skills as dynamic movement primitives [DMPs; 29, 30].

**Visual Keypoint-Conditioned Policies for Object Manipulation**. For the collaborative gift-bag assembly setting (§4.1), we implement skills $\pi_{\text{goto}}$ and $\pi_{\text{pickup}}$ via learned keypoint-conditioned models that ground object referring expressions (e.g., "a green toy car") to point clouds in a scene [31]. Specifically, given an (image, language) input, we first predict a 2D keypoint $(x, y)$ that corresponds to the centroid of the desired object in pixel space, then use off-the-shelf models [i.e., FastSAM; 32] to produce an object segmentation mask. Finally, we deproject this mask through our calibrated camera to obtain the object's point cloud to inform manipulation. To ensure consistency, we use a learned mask propagation model [XMem; 33] to maintain a dictionary mapping language expressions to existing object masks – this allows us to ensure that referring expressions such as "the green toy car" always refer to the same object instance for the duration of the interaction (as opposed to predictions changing over time, confusing users). We provide further detail in §B.3.

Note that we adopt this modular approach (predict keypoints from language, then extract a segmentation mask) for two reasons. First, we found learning a visual keypoints model to be extremely data efficient (requiring only a couple dozen examples) and reliable, significantly outperforming existing open-vocabulary detection and end-to-end models such as OWL-v2 [34, 35] as well as vision-language foundation models such as GPT-4V [36]; we quantify these results via offline evaluation in §B.3. Second, we found keypoints to be the right interface for interpretability and teaching: to specify new object, users need only "click" on the relevant part of the image via our GUI (§3.3).

**Learning Dynamic Movement Primitives from Demonstration**. For our stop-motion animation setting (§4.2), the robot is tasked with controlling the camera to execute different cinematic motions such as panning, tracking, or zooming (amongst others); due to the dynamic nature of these motions, we implement skills as (discrete) DMPs [29, 30], where users teach new motions by providing a *single* kinesthic demonstration; using DMPs allows us to generalize motions to novel start and goal positions, while also providing nice affordances for re-timing trajectories (e.g., speeding up a motion by a factor of 2x, or constraining that a motion executes in $K$ steps). Furthermore, as rolling out a DMP generates an entire trajectory, we can provide users with a visualization of the robot's intended path via our custom GUI (§3.3; visualized in Fig. 6). To learn a DMP, we divide a user's

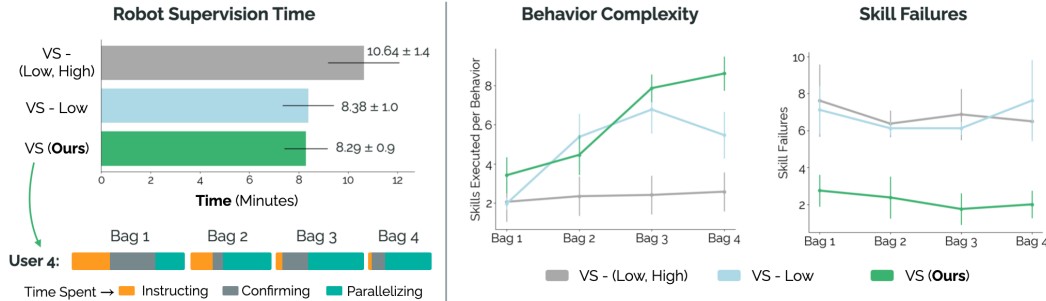

Figure 4: **User Study Quantitative Results**. We report robot supervision time **[Left]**, behavior complexity **[Middle]**, and skill failures **[Right]** across assembling individual gift bags in our user study (§4.1). Over time, users working with Vocal Sandbox (VS) systems teach more complex high-level behaviors, see fewer skill failures, and need to supervise the robot for shorter periods of time compared to baselines.

demonstration into a fixed sequence of waypoints, and learn a set of radial basis function weights (e.g., $\beta_{\mathsf{track}}$ from Fig. 1) that smoothly interpolate between them; we provide further detail in §B.4.

### 3.3 A Structured Graphical User Interface for Transparency and Intervention

Finally, we design a graphical user interface (GUI; Fig. 5) to provide users with *transparency* into the system state and to enable *targeted teaching*. Consider an utterance "pack the candy" and a failure mode that has the robot packing a different object (e.g., the ball) instead. Without any additional information, it is impossible for the user to identify whether this failure is a result of the LM planner (e.g., generating the incorrect plan pack(ball)), or a failure in the skill policy (e.g., incorrectly predicting a keypoint on the ball instead of the candy). The GUI counteracts this confusion by explicitly visualizing the plan and the interpretable traces produced by each skill (e.g., the predicted keypoints and segmentation mask, or the robot's intended path as output by a DMP) – these interpretable traces also double as interfaces for eliciting teaching feedback (e.g., "clicks" to designate keypoints). The GUI also provides 1) the transcribed user utterance, 2) the current "mode" (e.g., normal execution, teaching, etc.), and 3) the prior interaction history.

## 4 Experiments

We evaluate the usability, adaptability, and helpfulness of Vocal Sandbox systems through two experimental settings: 1) a real-world human-subjects study with $N = 8$ inexperienced participants for the task of collaborative gift bag assembly (§4.1), and 2) a more complex setting that pairs an experienced system-user (author of this work) and robot to film a stop-motion animation (§4.2). All studies were IRB-approved, with all participants providing informed consent.

### 4.1 User Studies: Collaborative Gift Bag Assembly

This study has participant work with the robot to assemble four gift bags with a fixed set of objects (candy, Play-Doh, and a toy car), along with a hand-written card (transcribed from a 96-word script). This task is repetitive and time-intensive, serving to study how well users can teach the robot maximally helpful behaviors to parallelize work and minimize their time spent supervising the robot.

**Participants and Procedure**. We conduct a within-subjects study with 8 non-expert user (3 female/5 male, ages $25.2 \pm 1.22$). Each user assembled four bags with three different systems, with a random ordering of methods across users. Prior to engaging with the robot, we gave users a sheet describing the robot's capabilities (i.e., the base API functionality), and instructions for using any teaching interfaces (if applicable). Prior to starting the bag assembly task, users were allowed to practice using each method for the unrelated task of throwing (disjoint) objects from a table into a trash can.

**Independent Variables – Robot Control Method**. We compare the full Vocal Sandbox system (**VS**) to two baselines. First, **VS - (Low, High)** (without Low, High), a *static* version of Vocal Sandbox that ablates both low-level skill teaching and high-level plan teaching, reflecting prior work like MOSAIC [8] that assume a fixed skill library and LM planner seeded with a base API. Our second baseline **VS - Low** ablates low-level skill teaching, but allows for teaching new high-level behaviors.

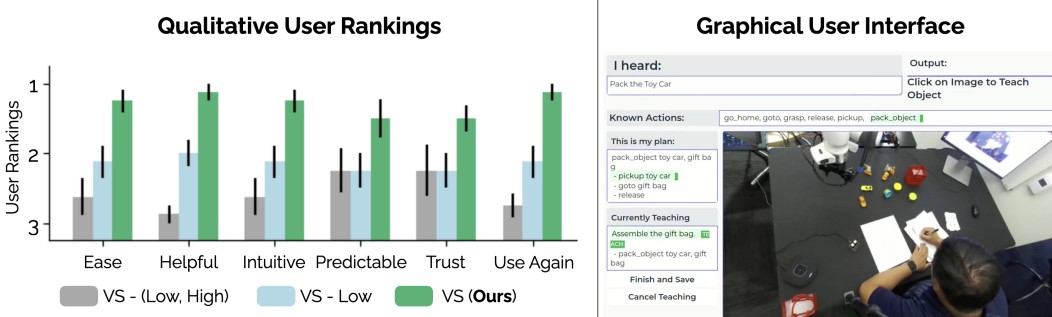

Figure 5: **User Study Subjective Results and System GUI**. We report qualitative user rankings for Vocal Sandbox (VS) and two baselines. With high significance ($p < 0.05$), we observe that VS outperforms the VS - (Low, High) baseline across all measures except predictability and trust **[Left]**. We also visualize the graphical user interface (GUI; §3.3) displayed to users when interacting with the system **[Right]**.

**Dependent Measures**. We consider objective metrics such as time supervising the robot, number of commands spoken, number of low-level skills executed per command (behavior complexity), as well as the number of teaching interactions at both the high- and low-level. Intuitively, we expect that methods capable of learning high-level behaviors from user feedback require less active supervision time and yield more complex behaviors than those using a static (naive) planner; similarly, we expect methods that permit teaching new low-level skills see fewer low-level skill execution errors. In addition to objective metrics, we asked users to fill out a qualitative survey after engaging with each method, describing their experience and ranking each method.

**Results – Objective Metrics**. We report objective results in Fig. 4. We measure robot supervision time (Fig. 4; Left) as a proxy for robot capability – a more capable robot should require less supervision (as it performs more actions autonomously). We observe that Vocal Sandbox (VS) systems outperform *both* the VS - (Low, High) and VS - Low baselines in terms of supervision time, demonstrating the capabilities afforded by the ability to teach at multiple levels of abstraction; the strengths of VS systems are made further evident in visualizing how users spend their time during a collaboration across instructing, confirming, and parallelizing (Fig. 4; Bottom-Left).

We additionally measure the complexity of (taught) high-level behaviors, measures as the number of low-level skills executed per language utterance (Fig. 4; Middle). We find that VS systems yield increasingly more complex behaviors compared to both baselines, with significantly ($p < 0.05$) more complex behaviors taught (and used) compared to the VS - (Low, High) baseline. This further highlights the importance of high-level teaching via structured function synthesis (§2.3). Of equal importance is the ability of the robot to learn low-level skills, exhibiting a declining rate of skill execution failures over time (Fig. 4; Right). We see that VS systems show significantly ($p < 0.05$) fewer skill failures than *both* baselines, stressing the isolated importance of teaching at the low-level. We further note that VS systems show fewer skill failures as a function of numbers of bags assembled, demonstrating the ability of VS systems to improve over time.

**Results – Subjective Metrics**. We report subjective survey results in Fig. 5 (Left), where users ranked the three methods across six different measures. In terms of ease, helpfulness, intuitiveness, and willingness to use again, we find that users prefer our VS system over both baselines due to its transparency and teachability, as users noted "teaching is useful" and "I loved how I was able to teach the robot certain skills". We also note that users significantly ($p < 0.05$) prefer VS to the naive VS - (Low, High) baseline, as it "felt chunkier to use". For predictability and trust, we observe that users prefer our VS system over baselines, however this trend is less pronounced because *any* autonomous execution incurs some loss of predictability (due to imperfect robot execution with learned policies).

## 4.2 Scaling Experiment: LEGO Stop Motion Animation

Finally, to push the limits of our framework, we consider a LEGO stop motion animation setting, where an experienced system-user (author of this work) works with the robot *over two hours of continuous collaboration* to shoot a stop-motion film. The user is the director, leading the creative vision for the film by directing footage and arranging LEGO set pieces, while the robot controls the

camera to capture different types of dynamic shots. Specifically, the user teaches the robot to perform cinematic concepts including "tracking", "zooming" and "panning" via kinesthetic demonstrations, which we use to fit different DMPs (as described in §3.2). The user iteratively builds on these behaviors to shoot progressively more complex frame sequences, ultimately resulting in a 52-second stop-motion film, consisting of 232 individual frames. Of the total frame count, 43% were shot with completely autonomous dynamic camera motions taught by the user – that is, camera motinos where the LEGO scene remained fixed, but the camera moved subject to a DMP rollout (e.g., a "zoom-in" shot). We found the taught skills were able to generalize across different start and end positions, as well as different timing constraints – for example "pan around slowly" commanding a pan_around motion with more frames ($N = 30$; 8 seconds), while "pan around quickly" modulated the same skill to execute in fewer frames ($N = 8$; 1.33 seconds). Over the two-hour long interaction, the robot executed on 40 novel commands – commands such as "let's frame the tower in this shot" to "zoom into Iron Man" (see Appx. D and our project page for more examples).

## 5 Related Work

Vocal Sandbox engages with a large body of work proposing systems for human-robot collaboration that pair task planning with with learned skill policies for executing actions; we provide an extended treatment of related work in §C.3, focusing here on prior work that center language models for task planning, or introduce generalizable approaches for learning skills from different feedback modalities.

**Task Planning with Language Models**. Recent work investigates methods for using LMs such as GPT-4 and PaLM [37–39] for general-purpose task planning [9, 16–18]. Especially relevant are approaches that use LMs to generate plans as programs [19, 40–43]. While some methods explore using LMs to generate new skills – e.g., by parameterizing reward functions [44, 45] – they require expensive simulation and offline learning. In contrast, Vocal Sandbox designs lightweight learning algorithms to learn new behaviors *online*, from natural user interactions.

**Learning Generalizable Skills from Mixed-Modality Feedback**. A litany of prior approaches in robotics study methods for interpreting diverse feedback modalities, from intent inference in HRI [8, 46, 47] to learning from implicit expressions [11, 48] or explicit gestures [12, 49, 50]. With the advent of LMs, language has become an increasingly popular interaction modality; however, most methods are limited to specific *types* of language feedback such as goal specifications [13, 51] or corrections [14, 52, 53]. In contrast, Vocal Sandbox demonstrates that language alone is not sufficient when teaching new behaviors – especially for teaching new object groundings or dynamic motions. Instead, our framework leverages *multiple* feedback modalities simultaneously to guide learning.

## 6 Discussion

We present Vocal Sandbox, a framework for situated human-robot collaboration that continually learns from mixed-modality interaction feedback in real-time. Vocal Sandbox has two components: a high-level language model planner, and a family of skill policies that ground plans to actions. This decomposition allows users to give targeted feedback at the correct level of abstraction, in the relevant modality. We evaluate a Vocal Sandbox system in a user study across $N = 8$ participants, and observe that our Vocal Sandbox system is preferred by users (+18.2%), requires minimal supervision (-22.1%) and yields more complex autonomous performance (+19.7%) with fewer failures (-67.1%) compared to non-adaptive baselines.

**Limitations and Future Work.** As execution relies on low-level skills that are quickly adapted from sparse feedback, this framework struggles in dexterous settings (e.g., assistive bathing) where more data is necessary to capture behavior nuances. Another shortcoming is that the collaborations enabled by our system are relatively homogeneous – users are teachers and robots are followers – which is not suited for all settings. Future work will explore algorithms for cross-user improvement as well as sample-efficient algorithms to learn more expressive skills. We will also further probe the user's model of robot capabilities to investigate questions about human-robot trust and collaboration styles.

**Acknowledgments**

This work was supported by the Toyota Research Institute (TRI), the DARPA Friction for Accountability in Conversational Transactions (FACT) Program, the AFOSR Young Investigator Program, the Cooperative AI Foundation, the NSF (Awards #1941722, #2006388, #2125511), the Office of Naval Research (ONR Award #N000142112298), and DARPA (Grant #W911NF2210214). Jennifer Grannen is grateful to supported by the NSF Graduate Research Fellowship Program (GRFP). Siddharth Karamcheti is grateful to be supported by the Open Philanthropy Project AI Fellowship. Any opinions, findings, and conclusions expressed in this article reflect those of the authors and do not necessarily reflect the views of the sponsoring agencies. Finally, we thank Yuchen Cui, Suneel Belkhale, David Hall, and our anonymous reviewers for their thoughtful feedback and suggestions.

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

# Overview

In the appendices below, we provide additional details around the implementation of the various components of *both* Vocal Sandbox systems (i.e., for both the collaborative gift-bag assembly and the LEGO stop-motion settings), including details around the rest of the system architecture (e.g., speech recognition, text-to-speech). We then extend our discussion from the main body of paper, building on additional opportunities provided by our framework for future research.

A large portion of our paper is grounded in user studies and extended interactions; videos of these studies can be found on our project website: https://vocal-sandbox.github.io

Furthermore, our work involves prompting GPT-3.5 Turbo with function calling [v11-06; 27, 28] through the public OpenAI API; we provide these prompts directly (as code) on our website as well: https://vocal-sandbox.github.io/#language-prompts

An overview of each appendix is as follows:

---

### Appx. A – *Motivating Questions*

We index a list of motivating questions that may arise from reading the main text and that we expand on further here (e.g., "why adopt a modular vs. an end-to-end approach?").

Our answers here are *direct*, linking to concrete sections further on in the appendices.

### Appx. B – *Implementing Vocal Sandbox*

We provide complete implementation details for both Vocal Sandbox systems we present in the main body of the paper; we additionally include details around the rest of the system architecture (e.g., speech-to-text, robot control, etc.):

#### §B.1 – *System Architecture*

Additional system details around implementing real-time speech recognition and text-to-speech, with latency and pricing statistics.

#### §B.2 – *Language Models for Task Planning and Skill Induction*

GPT-3.5 Turbo prompting details, generation hyperparameters, and latency statistics.

#### §B.3 – *Visual Keypoint-Conditioned Policy Implementation*

Additional implementation details and static evaluations for our visual keypoint-conditioned policy (for robust object manipulation).

#### §B.4 – *Learning Discrete Dynamic Movement Primitives from Demonstration*

Formalism of discrete dynamic movement primitives (DMP) and learning algorithm; additional details about the DMP features we employ (re-timing and goal editing).

#### §B.5 – *Physical Robot Platform & Controller Parameters*

Robot platform and controller implementation; ensuring compliant control and safety.

### Appx. C – *Extended Discussion & Future Work*

We provide an extended discussion on themes from the paper, from the importance of modularity and transparency in developing Vocal Sandbox, the broader impact of Vocal Sandbox in the context of human-robot collaboration, and directions for future work.

### Appx. D – *Additional Experiment Visualizations*

We provide additional visualizations of experiment settings and results, including stills from the LEGO stop motion interaction and final film product.

# A    Motivating Questions

**Q1**. *If I wanted to implement a Vocal Sandbox from scratch, what components would I need? How do the current experiments handle real-time speech-to-text and text-to-speech? What about pricing – what was the cost of running the Gift-Bag Assembly User Study ($N = 8$)?*

Beyond the language model task planner that uses GPT-3.5 Turbo with function calling [v11-06; 27, 28], and the (lightweight) learned skill policy, we use a combination of Whisper [54] for real-time speech recognition (mapping user utterances to text), and the OpenAI text-to-speech (TTS) API [55] for vocalizing confirmation prompts and querying users for teaching feedback. All models, cameras, and API calls are run through a single laptop equipped with an NVIDIA RTX 4080 GPU (12 GB).

For the gift-bag assembly user study ($N = 8$), the total cost of all external APIs (Whisper, OpenAI TTS, GPT-3.5 Turbo) amounted to \$0.47 + \$0.08 + \$1.24 = \$1.79. For the entirety of the project, GPT-3.5 API spend was \$5.79, with ∼\$4.00 spent on Whisper and TTS (< \$10.00 total).

**Q2**. *Given the use of powerful closed-source foundation models such as GPT-3.5/GPT-4, why adopt a modular approach for implementing the visual keypoints (and similarly dynamic movement primitives for learning policies)? Why not adopt an end-to-end approach building on top of GPT-4 with Vision, or existing pretrained multitask policies?*

We choose to adopt a modular approach in this work for two reasons. First, existing end-to-end models are still limited when it comes to fine-grained perception and grounding; we quantify this more explicitly through head-to-head static evaluations of our keypoint model vs. pretrained models such as OWLv2 [34, 35] in §B.3. Second, we argue that modularity allows users to systematically *isolate failures* and address them via multimodal feedback, at the right level of abstraction. We expand on this further in §C.1.

**Q3**. *The baseline methods in the user study (§4.1) are framed as ablations, rather than instances of existing systems that combine language model planning with learning from multiple modalities (e.g., on-the-fly language corrections, gestures). How were these ablations chosen? What is their explicit relationship to prior work?*

In our user studies, we constructed our baselines in a way that best represented the contributions of prior work while still fulfilling the necessary prerequisites to perform in our situated human-robot collaboration setting. Though we labeled these "ablations" in the paper, each one is representative of prior work – connections we make explicit in §C.2.

**Q4**. *How does Vocal Sandbox fit into the context of prior human-robot interaction works? What are the new capabilities Vocal Sandbox is bringing to the table?*

While the main body of the paper situates our framework against prior work in task planning and skill learning from different modalities, Vocal Sandbox builds on a rich history of work that develops systems for different modes of human-robot interaction. We provide an extended treatment of related work, as well as directions for future work in §C.3.

# B Implementing Vocal Sandbox

Implementing a system in the Vocal Sandbox framework requires not only the learned components for language-based task planning and low-level skill execution, but broader support for interfacing with users via automated speech-to-text, text-to-speech for vocalizing failures or confirmation prompts, as well as a screen for visualizing the graphical user interface. We describe these additional components, as well as provide more detail around the implementation of the learned components of the systems instantiated in our paper over the following sections.

## B.1 System Architecture

For robust and cheap automated speech recognition (mapping user utterances to text), we use Whisper [27, 54], accessed via the OpenAI API endpoint. Whisper is a state-of-the-art model meant for natural speech transcription, and we find that the latency for a given transcription request (< 0.5s round-trip) is more than enough for all of our use-cases. API pricing is also affordable, with the Whisper API (through OpenAI) charging $0.006 / minute of transcription (less than $0.50 to run our entire gift-bag assembly user study). Note that we implement speech-to-text via explicit "push-to-talk" interface, rather than an alternative "always-listening" approach; we find that this not only allows us to keep cost and word-error rate down, but improves user experience. By gating the listening and stop-listening features with explicit audio cues, users are more aware of what the system is doing, and can more quickly localize any failures stemming from malformed speech transcriptions.

In addition to automated speech-to-text, we adopt off-the-shelf solutions for real-time text-to-speech; this is mostly for implementing confirmation prompts ("does this plan look ok to you?") and for vocalizing the system state, but also includes an adaptive component when probing users to teach new visual concepts or behaviors ("I'm sorry, I'm not sure what the 'jelly-candy thing' looks like, could you teach me?"). For these queries, we use the OpenAI TTS API [55] with a similarly affordable pricing scheme of $15.00 per 1M characters (or approximately 200K words); to run our gift-bag assembly study, this cost fewer than $0.08. For hardware (for both speech recognition and text-to-speech), we use a standard USB speaker-microphone (the Anker PowerConf S3).

To visualize the graphical user interface to users, we use an external monitor (27 inches), placed outside of the robot's workspace. We drive the GUI, all API calls (speech recognition, text-to-speech, and language modeling via GPT-3.5 Turbo), ZED 2 camera, and all our learned models – including our visual keypoint-conditioned policy, FastSAM [32], and XMem [33] – from a single Alienware M16 laptop with an NVIDIA RTX 4080 GPU with 12 GB of VRAM; this laptop was purchased matching the DROID platform specification [26].

**Modifications for Gift-Bag Assembly User Study**. For the gift-bag assembly user study ($N = 8$) we implement the "push-to-talk" speech recognition interface with physical buttons placed on the table; users are provided two buttons – one for "talking" and one for "cancelling" the prior actions (which serves a dual function as a secondary, software-based emergency stop when the robot is moving). These buttons are placed on the side of the user's non-dominant hand, always within reach.

**Modifications for LEGO Stop-Motion Animation**. For the LEGO stop-motion animation study, we use the same components as above, with two additions. As the expert user is directing and framing individual camera shots during the course of the collaboration, they add an additional laptop (a MacBook, running Stop Motion Studio) to the workspace (disconnected from the rest of the system). As the user requires both hands free for this study (for articulating LEGO minifigures and structures, or editing the clip on their laptop), we replace the tabletop "push-to-talk" buttons with a USB-connected foot pedal with two switches with the same recognition and cancel functionality.

## B.2 Language Models for Task Planning and Skill Induction

As described in the main body of the paper, we use GPT-3.5 Turbo with function calling [v11-06; 27, 28] as our base language model for the task planner. This was the latest, most affordable, and highest latency language model at the time we began this work (prior to the release of GPT-4 and

**Comparing Visual Keypoint Models**

**Mujoco GUI for DMP Visualization**

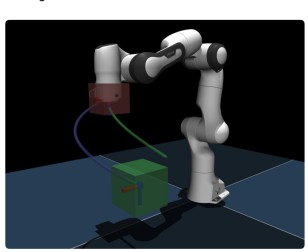

"yellow toy car"      "the play-doh"

■ OWLv2 (Ensemble)    ■ GPT-4V    ■ VS (**Ours**)

Figure 6: **Visual Keypoint Static Evaluation & DMP Visualizer**. We visualize visual keypoint predictions for object locations across a clean environment **[Left]** and a more difficult, cluttered environment **[Middle]**. We also highlight the trace generated by our Dynamic Movement Primitive (DMP) skills, rendered in a Mujoco [56] simulation environment to display the robot path before execution.

GPT-4o), with a response time between 1-3s on average, and a cost of \$2.00 / 1M tokens; for this work, the total cost we spent on GPT 3.5 API calls (including development) was \$5.79, with the gift-bag assembly user study itself only amounting to \$1.24 of the total spend.

We use the GPT-3.5 function calling capabilities throughout our work, which requires formatting our API specification following a custom JSON schema set by OpenAI; we provide these function calling prompts (and all GPT-3.5 prompts for our work) on our supplementary website for easy visualization: https://vocal-sandbox.github.io/#language-prompts. All language model outputs were generated with low-temperature sampling (0.2).

### B.3    Visual Keypoint-Conditioned Policy Implementation

As described in the main body of the paper, we use three components to implement Vocal Sandbox's object manipulation skills: (1) a learned language-conditioned keypoints model, (2) a pretrained mask propagatation model [XMem; 33], and (3) a point-conditioned segmentation model [FastSAM; 32].

Our learned keypoint model predicts object centroids from language, enabling us to generalize across object instances where XMem struggles. Given an RGB image $o_t \in \mathbb{R}^{H \times W \times 3}$ and natural language literal $c_{\text{ref}}$ from the high-level language planner, it predicts a matrix of per-pixel scores $\mathcal{H} \in [0, 1]^{H \times W}$. We take the coordinate-wise argmax of $\mathcal{H}$ as the predicted keypoint. We implement our model with a two-stream architecture following Shridhar et al. [57] that fuses pretrained CLIP [58] textual embeddings with a fully-convolutional architecture. We train this model on an small, cheap-to-collect dataset of 25 unique images each annotated with 3 keypoints (75 examples total). To fit our model, we create heatmaps from each ground-truth label, centering a 2D Gaussian around each keypoint with a fixed standard deviation of 6 pixels; we train our model by minimizing the binary cross-entropy between model predictions and these heatmaps, augmenting images with various label-preserving affine transformations (e.g., random crops, shears, rotations).

Our mask propagation model, XMem [33] tracks object segmentation masks from one image frame to the next; we provide a brief overview here. XMem is comprised of three convolutional networks (a query encoder $e$, a decoder $d$, and a value encoder $v$) and three memory modules (a short-term sensory memory, a working memory, and a long-term memory). For a given image $I_t$, the query encoder outputs a query $q = e(I_t)$ and performs attention-based memory reading from working and long-term memory stores to extract features $F_{c_{\text{ref}}}$, where $c_{\text{ref}}$ is the language utterance (e.g., "candy"). The decoder $d$ then takes as input $q$, $F$, and $h_{t-1}$ (the short-term sensory memory) to output a predicted mask $M_t$. Finally, the value encoder $v(I_t, M_t)$ outputs new features to be added to the memory history $h_t$. The query encoder $e$ and value encoder $v$ are instantiated with ResNet-50 and ResNet-18 [59] respectively. The decoder $d$ concatenates the short-term memory history $h_{t-1}$ with the extracted features $F$, upsampling by a factor of 2x until reaching a stride of 4. While upsampling, the decoder fuses skip connections from the query encoder $e$ at every level. The final feature map is

passed through a $3 \times 3$ convolution to output a single channel logit which is upsampled to the image size. See Cheng and Schwing [33] for additional details.

Finally, our point-conditioned segmentation model, FastSAM [32], is used to extract an object mask from a predicted keypoint. It has two components: a YOLOv8 [60] segmentation model $s$ for all-instance segmentation, and a point prompt-guided selection for identifying the object mask in which the point lies. From a given predicted keypoint $p$, the segmentation model outputs the mask $M$ from $s$ that encompasses $p$. We refer to [32] for additional details. This predicted mask $M$ is subsequently added to the XMem memory storage after being passed through the value encoder $v$.

Robot actions are coded as parameterized primitives (i.e., `pick_up` or `goto`) that take object locations as input and output trajectories.

**Static Evaluations – Robust Object Grounding**. To highlight the need for a data-efficient, domain-specific vision system, we evaluate the performance of our vision module implementation (as described above) compared to existing closed-source foundation models such as OWLv2 and GPT-4V. To compare, we consider an (image, annotation) dataset of all visual queries from the $N = 8$ user study, where the annotation is where the user confirmed or manually selected a correct object location. We report measures for accuracy and precision – keypoint mean squared error in pixel distance and success counts for predictions within a toy-car radius (14 pixels) from the annotation. For the Vocal Sandbox predicted mask, the centroid of the mask is used for these point-to-point calculations. We observe that while the mean squared error across all three methods are comparable, our Vocal Sandbox vision module greatly outperforms the foundation model baselines in the precision metric. This is because all the objects are clustered together on a table (Fig. 6) – randomly selecting between these objects yields low MSE predictions, however a nearby prediction is not sufficient to identify and isolate the correct object for grasping.

|  | Keypoint MSE (px) | Precision |
|---|---|---|
| OWLv2 Ensemble (ViT-L/14) | $35.3 \pm 1.01$ | $1.83 \pm 0.91$ |
| GPT-4-Turbo (w/ Vision) | $36.39 \pm 1.73$ | $15.94 \pm 2.55$ |
| Vocal Sandbox (Ours) | $30.46 \pm 3.61$ | $69.41 \pm 3.12$ |

### B.4 Learning Discrete Dynamic Movement Primitives from Demonstration

For our LEGO stop-motion animation setting, we implement our low-level skill policy as a library of discrete Dynamic Movement Primitives [29, 30]. We adopt the traditional discrete DMP formulation from Ijspeert et al. [30], defining a second-order point dynamical system in terms of the system state $y$, a goal $g$, and phase variable $x$ such that:

$$\tau \ddot{y} = \alpha_y(\gamma_y(g - y) - \dot{y}) + f(x, g); \qquad \tau \ddot{x} = -\alpha_x x$$

where $\alpha$ and $\gamma$ define gain terms, $\tau \in (0, 1]$ denotes a temporal scaling factor, and $f(x, g)$ is the *learned forcing function* that drives a DMP to follow a specific trajectory to the goal $g$; $f(x, g)$ is implemented as a learned linear combination of $J$ radial basis functions and the phase variable $x$ such that:

$$f(x, g) = \frac{\sum_{j=1}^{J} \psi_j w_j}{\sum_{j=1}^{J} \psi_j} x(g - y_0); \qquad \psi_j = \exp(-h_j(x - c_j)^2)$$

where $c_j$ and $h_j$ are the heuristically chosen centers and heights of the basis functions, respectively. We fit the DMP weights $\beta = \{w_1, w_2 \ldots w_J\}$ with locally-weighted regression (LWR) from the provided kinesthetic demonstration. For all DMPs in this work, we use $J = 32$, with gain values $\alpha_y = 25$, $\gamma_y = \frac{25}{4}$ and basis functions parameters set following prior work [30].

We choose (discrete) DMPs to implement skill learning as they permit efficient learning from a kinesthetic demonstration, and have two properties that enable rich generalization to 1) new goals (by

specifying a new $g$) and 2) arbitrary temporal scaling (by rescaling $\tau$). This lets us induce a simple algebra for parameterizing our policy $\pi_{d,\beta} : (c_{\text{ref}}, l, N)$, indexing each learned DMP with a learned referent $c_{\text{ref}}$, a new goal location $l$, and a number of waypoints $N$ (used to set $\tau$) – in other words, allowing us to learn a new DMP – track(loc: Location) – that we can call with arbitrary new locations (from novel initial states) with arbitrary timing parameters (e.g., "can you track around Loki in 30 frames" or "I need a tracking shot around the tower... let's try 2 seconds").

**Visualizing DMP Rollouts**. Another advantage of using DMPs for parameterizing control is that they allow us to visualize entire trajectories *prior to execution*. Similar to how we visualize the keypoints and object segmentation masks in the collaborative gift-bag assembly setting, we provide a GUI that shows the robot and the planned path (and end-effector poses) via a simple MuJoCo-based viewer. Fig. 6 (Right) provides an example – we plot the original kinesthetic demonstration relative to the current robot pose in green (for reference), and the planned DMP trajectory in blue, along with the end-effector orientation frames at the beginning and end of the trajectory. Users additionally can dynamically advance the simulation to visualize the entire rollout (at the actual speed of execution).

### B.5 Physical Robot Platform & Controller Parameters

We use a Franka Emika Panda 7-DoF robot arm with a Robotiq 2F-85 parallel jaw gripper following the platform specification from DROID [26]. The robot and its base are positioned at one side of a 3' x 5' table, across from the user, such that the user and robot share the tabletop workspace. We use an overhead ZED 2 RGB-D camera with known intrinsics and extrinsics. For robot control, we use a modified version of the DROID control stack based on Polymetis [61]. Low-level policies command joint positions at 10 Hz to a joint impedance controller which runs at 1 kHz. We implement two compliance modes: a *stiff* mode which is activated when the robot is executing a low-level skill, and a *compliant* mode for when the user provides a kinesthetic demonstration.

**Safety.** We include multiple safeguards to ensure user safety. Users have the option to cancel any proposed behavior when an interpretable trace is presented with a physical Cancel button as described in §B.1 – this prevents execution and immediately backtracks the Vocal Sandbox system. Second, during execution of any low-level skill, the user can interrupt the robot's motion with this button as well. This halts the robot's motion and it immediately becomes fully compliant. Lastly, during user studies, both the user and proctor have access to the hardware emergency stop button which cuts the robot's power supply and mechanically locks the robot arm.

## C   Extended Discussion & Future Work

The following sections expand on the discussion from the main body of the paper, with a specific focus on the benefits of Vocal Sandbox's modular design, before providing an extended treatment of our contributions and future directions in the broader context of systems for human-robot collaboration.

### C.1   On Modular vs. End-to-End Approaches

We develop Vocal Sandbox as a *modular* framework; the decoupled nature of the high-level language behavior planner from the low-level skill policies is explicit, and characterizes the rest of our contributions. Yet, this choice poses an important question – *why not an end-to-end approach*?

An initial answer stems from limitations in current models; our user studies show that "flat" planning with language models has several failure modes when it comes to reliability, while our static evaluations in §B.3 indicate deficiencies in ability for current multimodal models (e.g., GPT-4 Turbo with Vision) for high-precision language grounding in cluttered scenes. Yet, even if we consider a future where we have stronger end-to-end approaches that unify language, vision, and action [e.g., building on top of RT-2 or RT-H; 15, 62], we argue that modularity is an important feature in allowing users to isolate system failures and localize their feedback at the right level of abstraction.

Consider a common failure mode of end-to-end policies learned from data: visual robustness. Different degrees of distribution shift (e.g., introducing new distractor objects, or even perturbing the scene in small ways) not only hurt success rate [63], but they also affect the closed-loop execution in arbitrary ways [64], leading to suboptimal or unsafe trajectories. Worse is that errors only *cascade* as the robot or scene go further out of distribution, leading to even more unpredictable behavior.

Conversely, one of the more salient observations from our user study was how quickly users were able to not only identify failure modes in our system, but *co-adapt* to them. For example, within assembling the first two gift bags in the study, many users identified that the learned keypoint model was especially poor at predicting one category of object (Play-Doh). Rather than let this failure completely derail the task, users leaned on the modularity of our system to *isolate* this failure to the specific module (i.e., the visual keypoints-based policy) and sequence their feedback – teaching new high-level behaviors while *expecting* when and where the robot would fail. Specifically, users opted to teach a high-level behavior "`assemble_bag()`" that would always attempt to pack the Play-Doh into the gift bag as the final step, affording them the ability to maximally "disengage" from actively supervising the robot for the bulk of execution, only intervening at the last step. In other words, modularity in Vocal Sandbox gives users the leverage to to quickly understand the robot's capabilities, as well as the power to meaningfully build on its strengths, and adapt around its weaknesses.

### C.2   Explicit Connections Between System Ablations and Prior Work

Our user studies (§4.1) compares Vocal Sandbox systems to two baseline methods that we frame as different "ablations"; in this section, we explicitly link each of these ablations to prior work that builds systems for human-robot collaboration using language model planners, and that learn skills from different feedback modalities.

To make this link, we first want restate the contributions of Vocal Sandbox: 1) the ability to continually learn new high-level behaviors and low-level skills from mixed-modality teaching feedback, grounded in 2) a situated human-robot collaborative setting where we evaluate the ability of our system to provide more utility (assist the user) over time. The setting is important; the bulk of prior works frame human-robot collaboration as turn-taking between a human providing instructions or corrections from afar and a robot performing a task [12, 17, 51]. These works evaluate *binary success rate* – given user input and a task, did the robot succeed or fail.

In contrast, our setting assigns the robot and human clear roles within a collaboration (e.g., shoot a stop-motion film, with a robot controlling the camera and the human directing). Here, task success is a *necessary condition* – if the user could not direct the robot to perform a minimal set of actions, they would not use it (and instead perform the task completely on their own, or with a human partner). As

a result, we needed to develop a "baseline" system that ensured the robot remains a viable partner at every point during the collaboration. This is why we minimally define a set of primitives that cover the full space of robot capabilities needed, as well as any interfaces (e.g., the GUI) the user might need to teach. In other words, viability and recovery are prerequisites for all systems we evaluate. Given these prerequisites, we looked at our hypothesis – specifically that learning at both the high- and low-level through mixed modality feedback leads to more effective human-robot collaboration. We looked at prior works in collaborative task planning, approaches for handling on-the-fly language corrections, and methods for learning from other feedback modalities, identifying the following system "templates" to compare:

**1. "Static" Planning with Fixed Low-Level Skills**. These systems define human-robot collaboration as translation from user instructions to a fixed set of predefined skills. This system is reflective of the core contributions of recent work such as Text2Motion [17], ProgPrompt [18], and MOSAIC [8] – the latter of which we consider to be closest to ours.

Crucially, these works use LMs to map user intent to realizable robot plans specified as programs. We specifically build on MOSAIC, as we believe this system to be the most general – it does not assume access to full-state information a priori (i.e., the LM is not a priori prompted with all objects in the scene), instead invoking a learned perception model to localize objects if possible (as we do; §3.2). Combining this with our "prerequisite" system, **we get VS - (Low, High)** – a system reflective of MOSAIC that uses an LM to translate language input to skills from a fixed library, some of which rely on the output of a learned perception module.

**2. On-the-Fly Corrections: "Adaptive" Planning with Fixed Skills**. In works such as Inner Monologue [9], Code as Policies [19] and Yell at Your Robot [YAY; 14], robot systems go beyond static planning. Instead, they exhibit forms of reactivity: while the sets of skills in these works remain static, the planners incorporate user or environment feedback (as corrections, action feasibility, subtask success) to synthesize plans. Of these works, YAY adds an iterative component, explicitly adding structured behaviors to retrain a high-level policy in batches (similar to how we induce new high-level behaviors during teaching). **Adding this reactivity results in VS - Low** – a system that combines InnerMonologue with the iterative updates of YAY – a reactive LM planner that conditions on user corrections or other feedback to iteratively learn new high-level behaviors to guide planning.

**3. Iteratively Improving Low-Level Skills**. Finally, to explore the impact of adaptively growing our library of low-level skills learned from mixed-modality feedback we looked to prior work that learn skills from keypoint supervision as in KITE [31] and MOKA [65], gestures as in GIRAF [12], or as DMPs from kinesthetic demonstrations [30]. These works do not focus on collaborative multi-turn settings, instead proposing methods that are evaluated offline, given "static" input (e.g., evaluating success rate of a fully autonomous policy that is conditioned on gestures vs. not). As our contributions are focused more on the nature of collaboration when we allow users to teach new low-level skills, and not on the actual methods for skill learning, we use these methods as fixed components of our system. This results in our final Vocal Sandbox systems (VS) – where in our gift-bag assembly setting (§4.1, VS specifies skills via a learned visual keypoints model, and in the stop-motion setting (§4.2), VS specifies skills via DMPs. Note that for the keypoints model, we performed a static evaluation of different (pretrained) models in §B.3, finding our simple two-stream CLIP-based architecture to outperform existing open-vocabulary detectors like OWLv2 and foundation models like GPT-4V.

Structuring our ablations this way gives us a fair, apples-to-apples comparison against prior works such as MOSAIC (VS - Low, High), InnerMonologue and YAY (VS 0 Low), while demonstrating our novel contributions for enabling continual learning at both the high- and low-levels.

## C.3 Broader Context and Future Work

While the main body of our paper situates our framework against methods that use language models for task planning and learning low-level skills from multimodal feedback, Vocal Sandbox builds off a much larger body of work that build systems for different forms of human-robot interaction. The systems differ in the *modes* of collaboration they enable, from explicit human-robot teaming in

situated environments [1, 4, 6, 7, 66], to learned methods for shared autonomy [67–69], to platforms for assistive robotics [70–72], amongst many others [2].

While Vocal Sandbox is heavily inspired by this prior work, especially those that learn language interfaces for grounding user intent to low-level robot behavior [8, 73, 74], this is only the beginning. Future iterations of our framework will build on the types of interactions and learning we permit (e.g., multi-robot teaming or integrating modalities such as touch or nonverbal feedback), all driving towards general and seamless human-robot collaboration.

## D    Additional Experiment Visualizations

We present additional visualizations of the LEGO stop-motion animation experiment, including examples of user-taught camera motions and stills from the two hour interaction during filming.

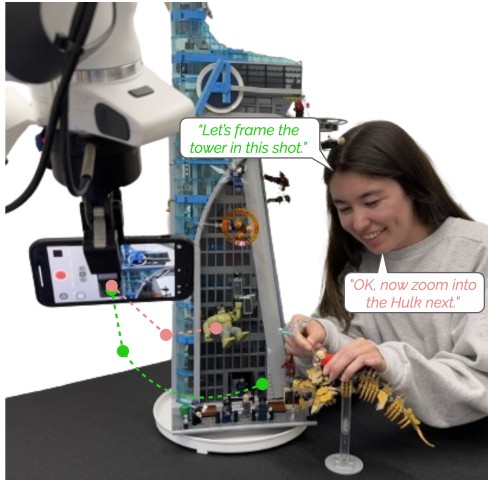

Figure 7: **LEGO User-Taught Camera Motions**. We present two examples of user-taught camera motions – "Let's frame the tower in this shot" and "OK, now zoom into the Hulk next" – along with their corresponding robot trajectories. Of the 232 frames shot for the film, 99 were shot with complex, user-taught camera motions such as the ones visualized here.

**Human-Robot Collaboration**  **LEGO Stop Motion Film**

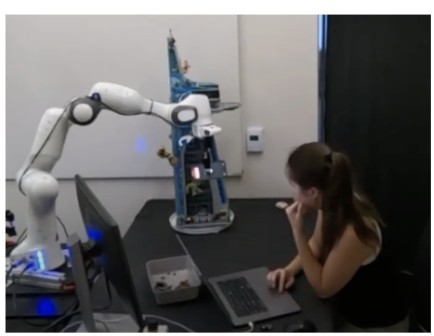
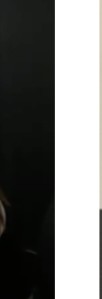
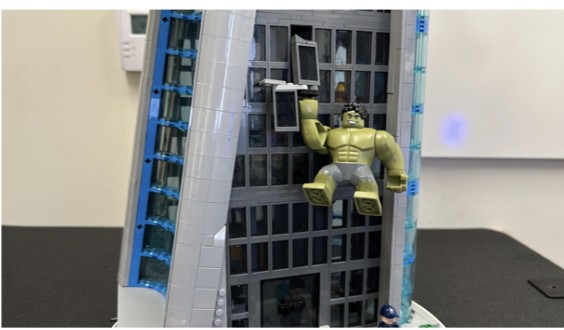

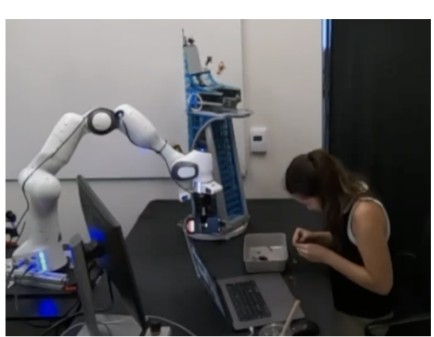
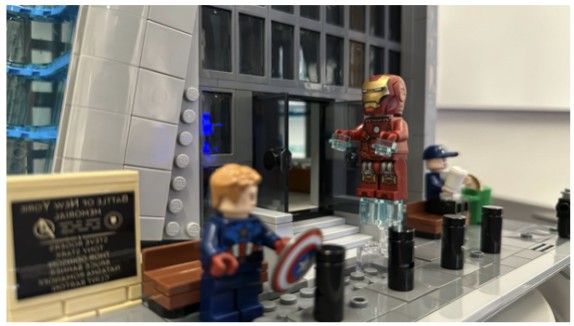

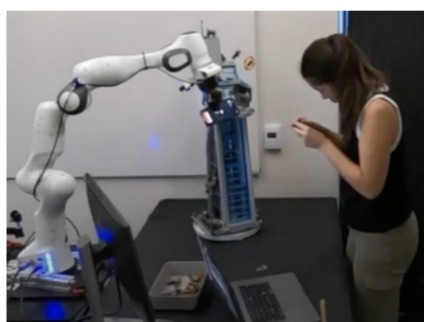
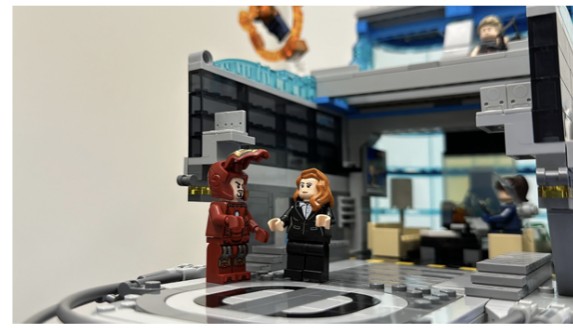

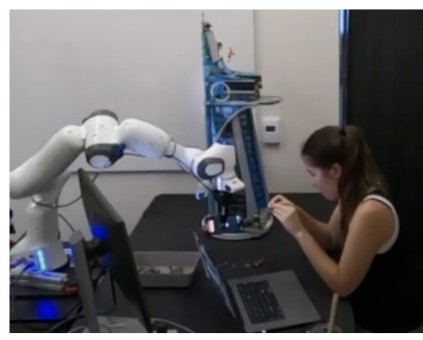
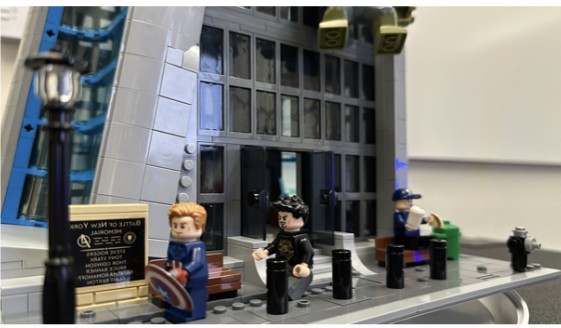

Figure 8: **Stills from LEGO Stop Motion Interaction**. We present stills from the two-hour continuous collaboration for shooting the LEGO stop motion animation [**Left**], as well as corresponding frames from the final film [**Right**]. The user directs and arranges LEGO figures while the robot controls the camera.

