# OpenReview forum: "Vocal Sandbox: Continual Learning and Adaptation for Situated Human-Robot Collaboration"
_robot-learning.org/CoRL/2024/Conference — CoRL 2024_

### Official Review · Reviewer_etUa · 2024-07-21
**Submission599**

**Originality:** 4
**Technical Quality:** 3
**Clarity Of Presentation:** 4
**Potential Impact:** 3
**Recommendation:** 3
**Confidence:** 2

**Review:**

Strengths

This is a very interesting paradigm and learning from online, multi-modal human interventions is an important research direction. The framework presented offers very promising system insights and results on favorable perceptions from novice users.

Weaknesses

The two baselines in this work are ablations of the system itself, each of which remove capabilities of the overall VS system. When the ability to correct skills and plans is removed, it is not surprising that users found the system easier to use, intuitive, and more performant. It would be effective to compare to existing works that combine learning from multiple modalities (ie. on-the-fly language corrections and learning from different feedback types).

Comments/Questions

How accurate is the skill trace? Is the learned robot skill demonstrated in simulation, and it would be great to show a snapshot of the GUI depicting how the robot trajectory looks while the trace is shown on screen when the user evaluates the trace. How often did the user disapprove of the skill trace? Is the user able to go back in and resume training if they disapprove of the skill?

What were the common failure modes when it comes to the ways users interacted with the system? Despite outperforming the baselines, did users struggle with any aspects of the VS system in particular. It would be excellent to elaborate qualitatively on why users found the system to be helpful and intuitive, and what parts of the system were not.

Why did VS outperform the VS (w/o Low, High) baseline on all measures except predictability and trust? Do you have any hypothesis as to why this is the case? Is the GUI driving predictability (via traces) and trust?

**Quality Of The Limitations Section:**

2

**Questions For Rebuttal:**

Co-adaptation is presented in the abstract as one of the main motivations of the visualization traces. What signs of co-adaptation were seen in the user study? How were users adjusting to the visualization traces? Did you notice shift in their teaching strategies over time?

Did users see a monotonic increase in performance as a result of further teaching? Where there instances where this was not the case?

Please see above section for further questions.

**Robotics Focus:**

4

**Summary Of Paper:**

Summary  This paper presents Vocal Sandbox, a framework for users to teach high-level behaviors through mixed-modality interactions. The framework enables users to give feedback and instruction through multiple modalities online during an collaboration. The Sandbox interface is over high-level behaviors and low-level skills. A high-level LLM planner maps language utterances to behavior plans, and learned low-level policies grounds these plans to robot actions. The low-level skills are trained through DMPs and keypoint-conditioned learned policies.  Online, during the interaction, the user gives the robot a natural language command (ie. track the hulk). The robot explains its failure, deferring to the user for choosing a next step. The user can either move on, or explicitly teach the robot what the new skill of tracking is. When teaching the new skill, the human defines a new behavior function, which can be composed of requisite skills. If a low-level skill is needed, the user can provide a demonstration. If a perceptual concept is needed, the user can label object segmentation masks. Finally, the system provides a visualization of the resultant robot trajectory.  Vocal Sandbox uses a interface for users to teach new skills, perceptual concepts, and visualize robot traces during teaching.  The sandbox is tested across two tasks, gift bag assembly across multiple non-expert participants, and stop motion animation with an expert user. Users strongly preferred the system over its ablated baselines in ease of user, helpfulness, and performance.

**Summary Of Recommendation:**

Overall, Vocal Sandbox presents a system for online behavior corrections to robots via multi-modal feedback. This is a very interesting paradigm and learning from online, multi-modal human interventions is an important research direction. The framework presented offers very promising system insights and results on favorable perceptions from novice users. Aspects of the evaluation and system performance can be further analyzed, but overall I recommend a weak accept.

---

### Official Review · Reviewer_nEiK · 2024-07-21
**Review of Submission 599**

**Originality:** 3
**Technical Quality:** 3
**Clarity Of Presentation:** 4
**Potential Impact:** 3
**Recommendation:** 3
**Confidence:** 4

**Review:**

Overall, this paper is very well-written, contains an impressive demonstration of a few combined techniques, and is validated via a user study.


Strengths
+ This paper combines several techniques well and allows for a new capability, improving human-robot collaboration.
+ The robot demonstration to create the 30-second stop-motion animation is impressive. It would be nice to attach this short film outside of the 3-minute as a standalone supplementary file.
+ This paper is well-written and contains sufficient information to understand the proposed approach.




Weaknesses
- There has been prior work (https://www.roboticsproceedings.org/rss18/p028.pdf) in humans teaching abstractions to robots via learning from demonstration. This paper's results display that humans have difficulty teaching robots different skills and behavior without oversight. Could you comment on why/how your approach and experiment lead to this capability?
- The term "expert" is used to describe the collaborator for the stop motion animation. As this "expert" is an author of this work, I would recommend grounding the expertise needed to perform this type of human-robot collaboration. Could you also comment on the expertise of the users in the human-subject study? Were these graduate students studying robotics and if so, are these results limited to this set of users?

Post-Rebuttal - The author's rebuttal was well-written and comprehensive. I appreciate the careful description regarding VS's relation to the results in Gopalan et al., the clarification regarding user expertise, and the anecdote about a user's experience with the proposed system. After reading all the reviews and their respective replies (and further discussions), I decided to maintain my score of Weak Accept.

**Quality Of The Limitations Section:**

3

**Questions For Rebuttal:**

- Please address the questions in the weaknesses section of the main review.

Further Questions:
- How do you ensure the learned low-level skills generalize well to other objects?
- Did users within the human-subject study need to reteach skills at any point or delete learned skills for any reason?

**Robotics Focus:**

4

**Summary Of Paper:**

This paper Vocal Sandbox (abbreviated as VS), a framework where users can teach both new high-level planning behaviors through spoken dialogue and provide teaching feedback to learn new low-level skills in real-time. The authors evaluate this framework in a user study with 8 non-expert participants and find that the system outperforms baselines both objectively and subjectively. Finally, the authors conclude by showing a human-robot collaboration example where the human and robot create a 30-second stop motion animation, an impressive feat.

**Summary Of Recommendation:**

Overall, this paper is very well-written, contains an impressive demonstration of a few combined techniques, and is validated via a user study. The contribution is incremental and a step forward for HRI, combining different techniques into a system for human-robot collaboration.

---

### Official Review · Reviewer_gPa9 · 2024-07-23

**Originality:** 3
**Technical Quality:** 3
**Clarity Of Presentation:** 3
**Potential Impact:** 3
**Recommendation:** 4
**Confidence:** 4

**Review:**

Strengths:
* Developing human-robot interaction frameworks which allow continuous adaptation are very important for fluid collaboration. Even in the era of large data-based foundation models, we can't expect pre-trained policies/frameworks to have perfect performance in all tasks that may be encountered in real world settings. I think the approach taken here which attempts to incorporate multiple feedback modalities across multiple levels of abstraction is very promising.

* The paper is easy to read and the provided examples are fairly intuitive. Given the complexity of the framework though, I feel like it might benefit from some sort of flow chart-based figure that indicates all the options available to a user to provide feedback.

Weaknesses:
* Aspects of the methodology are unclear and so it is hard to tell how useful this framework is in practice. For example, in Sec. 2.1 it is stated that the language model infers the parameters of skills being taught. But it's not clear a) whether the user can correct/override the proposed parameters, b) how these parameters are specified when providing a language instruction (does the user need to say "pan around slowly with N=30 frames" or does the language model infer parameters?), and c) how a user can correct parameters if they are inferred and are incorrect. Similarly, can a user provide corrections to behaviors purely via language or do they always have to do so manually through the GUI?\
What I am trying to get at is that it's unclear to me how much utility the framework (as proposed) offers as a whole as it's not clear how much user involvement is required at each step. For example, at risk of being overly reductive, is this a glorified GUI which simply exposes convenient utilities for training additional low-level skills and exposing them to a language model-based planner or is there something more useful/novel here?

* I think the paper would benefit from further elaboration/analysis on the complexity of user-defined behaviors. For example, Fig. 4 shows the complexity of behaviors increasing as users perform more tasks, but what do these look like? I am curious about things like how easy it is to modify existing behaviors. For example, if I am assembling gift bags using the behavior in Fig. 3, can I give an instruction like "Assemble another gift bag but also pack the book this time" which then adds another pack_object behavior to the end? Or does this require user intervention with the creation of a new behavior?

* For the amount of emphasis placed on mixed-modality feedback, there is very little discussion on how well it works. I don't see any analysis on feedback modalities in the experimental results section. Do users prefer one modality over another? When is each modality preferred? Is there value in even providing multiple modalities, as opposed to existing approaches which only use one? Does this depend on the underlying model/task and the type of mistakes that might occur?

**Quality Of The Limitations Section:**

3

**Questions For Rebuttal:**

1. Can the user correct/override the proposed skill parameters?

2. How are these parameters specified when providing a language instruction (does the user need to say "pan around slowly with N=30 frames" or does the language model infer parameters)? How can a user correct parameters if they are inferred and are incorrect?

3. Can a user provide corrections to behaviors purely via language or do they always have to do so manually through the GUI? I am wondering whether language is a feedback modality or simply used only for instructions (and therefore to define new skills/behaviors).

4. What do complex user-defined behaviors look like? Can a user modify existing behaviors (see example above with gift bag)?

5. Do users prefer one modality over another? When is each modality preferred? Is there value in even providing multiple modalities, as opposed to existing approaches which only use one? Does this depend on the underlying model/task and the type of mistakes that might occur?

6. Can a user modify an existing skill? For example, let's suppose that the user already taught the robot the "camera pan" skill. Can the user modify this skill to either add an additional parameter or create some sort of variation of it? Or do they have to create a completely new skill from scratch?

7. Does "(w/o Low, High)" in the experiments indicate that indicate that users can teach neither high-level behaviors nor low-level skills? The text states high-level only but that seems odd given the name.

**Robotics Focus:**

4

**Summary Of Paper:**

This paper introduces a framework for human-robot interaction that allows a user to continuously provide corrective and instructional feedback across multiple modalities and hierarchies. The proposed framework allows a user to teach both low-level skills, which can be taught via visual keypoints (conditioned policy) or kinesthetic demonstrations (DMP), and high-level behaviors taught via language. This framework also allows for corrective feedback in the event the system's plan does not match user expectations. The framework is evaluated through a user study in which the robot assembles a gift bag where it out-performs ablated methods in both subjective and objective metrics.

**Summary Of Recommendation:**

I think this paper is interesting and addresses a timely and relevant problem, but I do have some concerns about the methodology itself and am unsure how useful the proposed framework actually is.

---

### Decision · Program_Chairs · 2024-09-04

**Decision:**

Accept

**Comment:**

**Paper summary**

This paper contributes a framework for users to provide mixed-modality feedback (language, demonstrations, and labels) to robots in order to learn/improve both high-level and low-level behaviors. The paper contributes methods for integrating multi-modal feedback and providing transparency to the user about its behavior.

**Review summary**

Summary of strengths:
+ The paper addresses a key problem for human-robot collaboration: adaptation based on human feedback.
+ The proposed method is evaluated with a user study with non-experts.
+ The proposed method makes uses of multi-modal interactions, and allows a robot to learn both low-level and high-level information about a task.

Summary of weaknesses:
- More detail is needed about how users can (and cannot) interact with the interface at each step. Furthermore, the reviewers have noted several opportunities to highlight the users' experiences with system failures, approving/disapproving the robot's behavior, and preferences between modalities. **[Edit: The rebuttal now includes more details about the interface and user comments.]**
- The evaluation is missing comparisons to non-ablation baselines. **[Edit: The rebuttal clarifies that the ablated versions of the system are representative of SOTA baselines.]**
- The evaluation is missing detail about the user study participants and their familiarity with robots. **[Edit: The rebuttal clarifies the backgrounds of the participants.]**


**Response to rebuttal**

The authors all maintain their positive rating, and one reviewer has increased their rating from weak to strong accept. The reviewers agree that the rebuttal addresses their concerns. However, the clarifications have not yet been integrated into a revision.